# The Role of Chemical Modifications in the Genome of Negative-Sense RNA Viruses on the Innate Immune Response

**DOI:** 10.3390/v17060795

**Published:** 2025-05-30

**Authors:** María-Alejandra Ceballos, Mónica L. Acevedo

**Affiliations:** 1Laboratory of Molecular and Cellular Virology, Virology Program, Faculty of Medicine, Institute of Biomedical Sciences, Universidad de Chile, Santiago 8330015, Chile; maria.ceballos@ug.uchile.cl; 2Millennium Institute in Immunology and Immunotherapy, Santiago 8380453, Chile

**Keywords:** negative-polarity single-stranded RNA viruses, innate immunity, RNA chemical modifications, epitranscriptomic, N6-methyladenosine, N6,2′O-dimethyladenosine, N7-methylguanosine, 2′-O-methylation, inosine, pseudouridine

## Abstract

Negative-sense RNA viruses comprise a wide array of viral families, such as Orthomyxoviridae, Paramyxoviridae, Rhabdoviridae, and Morbillivirus, all of which are adept at inciting significant epidemic outbreaks. Throughout their replication cycle, these viruses engage in a variety of RNA modifications, during both the co-transcriptional and post-transcriptional phases, which are mediated by specific enzymatic activities. These chemical alterations play a critical role in shaping viral fitness, particularly in terms of evading innate immune responses. Key chemical modifications, such as adenosine methylation, 2′-O methylation of nucleosides, and adenosine-to-inosine editing, play critical roles in determining the stability, translational efficiency, and immune recognition of viral RNA. These modifications can reduce the activation of immune sensors, thereby suppressing interferon production and broader antiviral responses. In contrast, certain modifications may enhance immune recognition, which opens avenues for novel vaccine and antiviral strategy development. A comprehensive understanding of these RNA chemical modifications and their implications for virus–host interactions is essential for advancing therapeutic strategies aimed at manipulating innate immunity and optimizing the efficacy of RNA-based vaccines. This review examines the mechanisms and implications of RNA chemical modifications in negative-sense RNA viruses, emphasizing their dual roles in either evading or activating the innate immune system.

## 1. Introduction

Negative-sense RNA viruses encompass a diverse group that can lead to various human and animal infections. This group includes segmented genome viruses such as *Orthomyxiviridae*, which contains the influenza virus, and *Bunyaviridae*, including Hantavirus. Additionally, the non-segmented genome viruses found in *Rhabdoviridae* (e.g., rabies and vesicular stomatitis virus), *Paramyxoviridae* (e.g., measles, parainfluenza, and Sendai viruses), and *Pneumoviridae* (covering respiratory syncytial virus and metapneumovirus), are also included. These viruses rely on RNA-dependent RNA polymerase during replication to synthesize viral mRNA and produce subgenomic RNA of positive polarity, serving as a template for genomic RNA synthesis [1]. Consequently, these various viral RNAs engage different cellular networks and immune mediators that modulate the viral cycle [2,3]. In this light, chemical modifications of RNA can profoundly impact viral replication, innate evasion or evasion, and overall viral fitness.

Epitranscriptomic modifications of RNA represent a pivotal layer of post-transcriptional regulation, intricately influencing gene expression across various stages of RNA metabolism. These modifications, which can be enzymatically introduced or removed, play a crucial role in shaping virus–host dynamics and the broader regulatory framework of gene expression. To date, over 150 distinct RNA modifications with diverse chemical and structural characteristics have been characterized in both canonical and non-canonical nucleotides. Prominent modifications include pseudouridine (Ψ) and inosine (I). Additionally, a host of modifications has been identified in mRNAs and non-coding RNAs, such as N1-methyladenosine (m1A), N6-methyladenosine (m6A), 5-methylcytidine (m5C), N6,2-O-dimethyladenosine (m6Am), N7-methylguanosine (m7G) cap modifications, and 2′O-methylation, among others. Many of these modifications are intricately linked to the immune response and can significantly influence viral fitness.

The immune response to viral infections is orchestrated through complex interactions influenced by viral RNA modifications that affect host detection and response mechanisms. The innate immune system acts as the primary barrier, utilizing pathogen recognition receptors (PRRs) to identify viral RNA as pathogen-associated molecular patterns (PAMPs) [4]. Key PRRs include Toll-like receptors (TLRs), RIG-I-like receptors (RLRs), and NOD-like receptors (NLRs), with RIG-I and MDA5 tasked with RNA viruses detection, signaling via mitochondrial antiviral signaling protein (MAVS) [5]. This activation cascades through TBK1 and IκBɛ kinase, leading to the transcriptional activation of NFκB and interferon regulatory factors IRF3 and IRF7 [6]. Endosomal TLRs play a vital role as well; TLR3 engages TRIF and TLR7/8 interfaces with MyD88, propelling similar signaling pathways forward. The resultant transcription factors promote the production of type I interferons (IFN-α/β) [7,8,9,10,11] and cytokines, which bind to type I IFN receptors, subsequently inducing a suite of antiviral interferon-stimulated genes (ISGs) [12,13,14].

This review aims to critically examine the influence of chemical modifications in negative-sense RNA viruses, highlighting their implications across different phases of the replicative cycle and the innate immune response. Delving into the interplay between RNA modifications and immune mechanisms will provide vital insights into the pathogenesis of these infections.

## 2. N-6 Methyladenosine

The most abundant internal RNA modification is N-6 methyladenosine (m6A) (Figure 1), which has been described as regulating multiple cellular processes like stress response, microRNA biogenesis, cellular differentiation, and cancer, among others. The cellular machinery responsible for introducing m6A in RNA includes protein writers, erasers, and readers of m6A. The methyltransferases or writers are a complex consisting of Methyltransferase Like 3 (METTL3) [15], Methyltransferase Like 14 (METTL14) [16], and Wilms Tumor 1 Associated Protein (WTAP) [17]. METTL3 is the catalytic subunit, and METTL14 binds to RNA [18]. The methyltransferase complex adds a methyl group at the N6 position of adenosine. The demethylases or erasers comprise fat mass and obesity-associated protein (FTO) [19] and α-ketoglutarate-dependent dioxygenase alkB homolog 5 (ALKBH5) [20]. The main readers include the nuclear and cytoplasmic protein member, YTH family protein [21], and are responsible for exerting the functions of m6A. The distribution of m6A modification occurs in the consensus sequence DRACH [21]. However, m6A is preferentially located in the 5′- and 3′-untranslated regions (UTR) and near to stop codons. Depending on the cellular context and their position in the mRNA, the m6A modifications may favor the recruitment of splicing factors favoring alternative splicing and promote translation but may also induce the destabilization of the m6A-containing mRNA or its accumulation in stress granules. In this sense, the m6A modifications can exert their function on the mRNA metabolism by altering the interaction with m6A-regulatory or m6A-binding proteins or through an alteration of the RNA structure [22].

Although the methyltransferase complex is found in the nucleus, viral RNAs containing m6A modifications have been described in viruses that replicate in the nucleus and those that do it only in the cytoplasm. Indeed, it has been suggested that under certain conditions, the methyltransferase complex can translocate from the nucleus to the cytoplasm [23]. The m6A modification has been identified in the negative single-strand RNA of various viruses [24,25,26,27,28,29], and these modifications can exert both positive and negative effects on RNA replication and the assembly of viral RNA [30]. Notably, m6A modifications in viral RNA aid in the spread of different viruses by preventing early detection by the innate immune system. Typically, RNA viruses are recognized by RIG-I, a receptor found in the cytosol of host cells, and through membrane recognition patterns known as Toll-like receptors. These receptors activate innate immune signaling. However, when viral mRNAs—such as those from influenza [31] or respiratory syncytial virus (RSV) [32]—contain m6A-modified nucleotides, the strength of these signaling responses is diminished. This modulation affects the ability of the innate immune system to differentiate between cellular RNA and viral RNA [33,34]. Furthermore, it has been described that METTL3 needs to be phosphorylated by the type I interferon regulatory kinase TBK1 to promote the selectivity of METTL3 and, consequently, the translation of immune factors such as ISGs and IRF3 mRNAs [35].

The first negative single-stand RNA virus to be described with the m6A modification in its genome and mRNA is the influenza virus [36]. It was the first virus identified to express mRNA with internal m6A modifications, with approximately twenty-four such modifications across its various mRNAs. Among these, the most notable finding was that eight modified sites were identified in the mRNA segment encoding the hemagglutinin (HA) glycoprotein found on the surface of the virus [36]. Furthermore, research by Courtney et al. demonstrated that globally inhibiting the addition of m6A leads to a decrease in the expression and replication of influenza virus genes [31]. On the other hand, overexpressing the cellular m6A “reader” protein YTHDF2 enhances the expression and replication of the influenza virus. This suggests that m6A modifications boost HA expression during infection, ultimately promoting viral replication and pathogenesis.

In addition, respiratory syncytial virus (RSV) undergoes internal methylation through the addition of m6A at specific sites within several of its viral genes, including NS2, N, P, M, G, and L [32]. This modification plays a significant role in regulating the replication and expression of the virus [37,38,39]. Moreover, the removal of m6A methyltransferases has been found to increase the pathogenesis and virulence of RSV in vivo [32]. Notably, the mRNA of the viral G gene is characterized by a high concentration of m6A modifications [32]. The G glycoprotein is essential for the binding of RSV to host cells and has a considerable impact on innate immune responses. The presence of m6A modifications in the G gene mRNA enhances its stability, which, in turn, promotes increased translation. Furthermore, RSV strains with mutations at the m6A sites within the G gene exhibit reduced expression of the G glycoprotein and lower levels of other viral proteins, such as the N and F proteins.

Otherwise, analysis of differential host transcript expression following HRSV infection has been conducted using m6A-seq in both A549 and HeLa cells [32]. This analysis revealed a set of genes associated with the innate immune response that are upregulated in a conserved manner across both cell lines, including CXCL10, MDA5, OASL, and STAT1. Conversely, certain genes such as IRF3, Jak3, and TLR4 exhibited upregulation specific to individual cell lines, highlighting differential regulatory mechanisms in response to HRSV infection (Figure 2).

On the other hand, m6A-deficient RSV leads to increased RIG-I mediated type I IFN and lower levels of IL-6 and TNFα in the lungs, both of which are associated with lung inflammation [25]. Similarly, other near m6A-deficient viruses like Sendai virus (SeV), Measles virus (MeV), human metapneumovirus (HMPV), and vesicular stomatitis virus (VSV) show a similar profile in inducing a strong IFN response in a RIG-I-dependent manner [24,27].

The evidence strongly indicates that m6A is a crucial molecular marker for immune responses, frequently employed by negative single-stranded RNA viruses. Consequently, it is a vital component for evaluating viral replication in cellular, animal, and human models.

## 3. N6,2′-O-Dimethyladenosine

An additional modification identified in negative single-strand RNA viruses is N6,2′-O-dimethyl adenosine (m6Am), which has been found adjacent to the cap structure when the first or sometimes the second [40] transcribed nucleotide is an adenine [36,41]. This reversible modification involves adenosine methylation at the 2′-hydroxyl group, yielding 2′-O-methyladenosine (Am). In addition, adenosine can undergo additional methylation at the N6 position of Am, producing N6,2′-dimethyladenosine (m6Am). Both m6Am and N6-methyladenosine (m6A) arise through similar biochemical pathways: methylation at the N6 position of adenosine leads to the formation of m6A, while methylation of Am results in m6Am [42]. Nevertheless, m6Am exhibits several unique characteristics that distinguish it from m6A.

The modification m6Am is mediated by the host RNA polymerase II-associated phosphorylated CTD interacting factor 1 (PCIF1), which facilitates its addition. In contrast, the erasure of m6Am is carried out by the demethylase FTO [40,43,44]. The exact role of m6Am in viral RNA dynamics remains to be fully elucidated. However, it seems that m6Am may play a role in stabilizing viral RNAs similarly to the mechanisms observed in host mRNAs [45]. This implies that viruses may exploit this modification mechanism to regulate the intracellular concentrations of viral RNA. In the case of vesicular stomatitis virus (VSV), as well as the rabies and measles viruses, the modification of viral mRNA is mediated by the host enzyme PCIF1 [46]. When cells deficient in PCIF1 were treated with Interferon-ß, there was a significant suppression of both gene expression and VSV replication. This attenuation was found to be dependent on the activity of PCIF1 [46]. Furthermore, exploring the ramifications of this modification on the RNA of other viral pathogens will be essential for a comprehensive understanding of its effects on replication and immune response. Notably, PCIF1 has been implicated in pro-inflammatory responses, a function that remains to be thoroughly evaluated within the context of viral infections [47].

## 4. Inosine

Inosine (I) is a common RNA modification that occurs in double-stranded RNA regions through the deamination of the C6 position of adenine (A), leading to the synthesis of inosine [48,49,50,51]. This editing process is facilitated by RNA-specific adenosine deaminases (ADARs) [52]. In humans, the ADAR protein family comprises three members: ADAR-1, ADAR-2, and ADAR-3. Notably, ADAR-1 exists in two isoforms in mammals: the longer, interferon-inducible p150 isoform, which is predominantly found in the cytoplasm, and the shorter p110 isoform, which is constitutively expressed and primarily located in the nucleus [53]. Consequently, viruses that replicate within the cytoplasmic environment can enhance the functional activity of the ADAR-1 p150 isoform [54]. Functionally, inosine behaves similarly to guanosine (G), and the conversion of adenosine to inosine can significantly influence biological processes. This modification can affect the ribosome coding capacities, modify RNA polymerase activity, create or obliterate splicing sites, and alter RNA conformational dynamics [50,55,56].

Notably, it has a preferential base pairing with cytosine, which serves as a crucial marker for identifying ADAR editing sites; genomic adenines are interpreted as guanines in the corresponding complementary DNA sequences [57]. The deamination process results in the formation of an unstable inosine–uracil (I-U) base pair, which can induce structural and stability changes within the deaminated RNA duplex [51,58]. Furthermore, additional modifications of viral RNA mediated by ADAR have been characterized, where the edited RNA undergoes reverse transcription, allowing for inosine pairing with cytosine, thus replacing adenine with guanine at the edited site [59]. Inosine has been identified in various types of viral RNA, including respiratory syncytial virus (RSV), influenza virus, and parainfluenza virus [60,61,62].

During viral infections, ADAR-1 operates alongside a variety of interferon-stimulated proteins, playing a crucial role in modulating viral replication and infection dynamics [63,64,65]. The presence of inosine in double-stranded regions of viral RNA has significant implications for the coding sequences of viral proteins. ADAR-1 is thought to exert its antiviral effect by introducing mutations within the viral genome, thereby altering the functionality of viral proteins and potentially impairing viral replication [58].

Among the viruses where this modification has been described is RSV, where, during replication, the single-stranded negative-sense RNA is transcribed into double-stranded RNA by an RNA-dependent RNA polymerase [60]. ADAR-1 then edits this double-stranded RNA, incorporating inosine, which aids in the detection of non-self RNA by innate immune sensors [66]. Consequently, the modified RNA is recognized by Toll-like receptors (TLRs), triggering the production of antiviral cytokines, such as IL-6, IFN-β, and TNF-α [67]. This recognition activates mitogen-activated protein kinase (MAPK) pathways, particularly through the TLR-3 receptor [67]. Changes in the secondary structures of inosine-associated viral RNA are thought to be detected by TLR7 and TLR8 as well [68].

In the case of the measles virus, hypermutations from adenine to guanine catalyzed by ADAR-1 have been observed in the viral RNA [69]. The ability of ADAR-1 to edit the genomes of paramyxoviruses is particularly notable, given that viral RNA is closely associated with nucleocapsid proteins [61]. Therefore, the double-stranded RNA structures recognized and edited by ADAR-1 are likely limited [61]. Defective genomes can emerge during RNA replication if the polymerase prematurely terminates synthesis, potentially leading to a hybrid of genome and antigenome known as defective interfering RNAs (DI-RNAs) [70]. Transitions such as A-to-G or U-to-C have been identified within these DI-RNAs [71].

Further evidence has been gathered from the parainfluenza virus, part of the *Paramyxoviridae* family, where induction of errors, specifically U-to-C transitions, occurs in the viral genome. The biased hypermutation appears to be an activity analogous to that of the ADAR enzyme [62]. Inosine modifications have also been associated with the influenza virus [72], although the exact role these changes play in the viral replicative cycle remains unknown. Consequently, further research is necessary to elucidate the role of ADAR and the underlying mechanisms of viral biased hypermutation, as well as to investigate the implications of these hypermutations on the innate immune response.

## 5. Pseudouridine

Pseudouridine (Ψ, 5-ribosyluracil) is one of the most prevalent post-transcriptional modifications found in various noncoding RNAs, including transfer RNA (tRNA), ribosomal RNA (rRNA), and messenger RNA (mRNA) [73,74,75]. This modification results from the isomerization of uridine, a process primarily catalyzed by ribonucleoprotein complex H/ACA, which acts as an RNA pseudouridylase [76,77]. Additionally, pseudouridine synthases (PUSs) play a significant role in this irreversible modification [78,79].

Despite the extensive presence of pseudouridine in cellular RNAs, studies focusing on its occurrence in negative-sense RNA virus are scarce, and the role of this modification in the regulation of viral gene expression remains largely unexplored. Notably, pseudouridine modification has been characterized in the influenza virus through RNA immunoprecipitation assays utilizing specific antibodies against pseudouridine [72].

Moreover, emerging evidence indicates that external stressors can influence the levels of pseudouridine [80,81], suggesting that this modification may be inducible and potentially linked to the stress response mechanisms activated during viral infections. In other viral models, modified viral RNA has demonstrated the ability to evade detection by host immune sensors such as RIG-I; RNAs containing pseudouridine exhibit a high affinity for RIG-I but do not elicit the typical conformational shifts associated with robust innate immune signaling [33].

While it remains uncertain whether this is a universal mechanism applicable to all viruses, it raises important questions about the functional significance of pseudouridine in negative-sense RNA viruses and warrants further investigation to elucidate its role in viral pathogenesis and immune evasion.

## 6. Viral Methyltransferases, Direct Addition of N7-Methylguanosine (m7G), and 2′-O-Methylated Nucleotides

Cellular mRNAs possess modified nucleotides at their 5′-end, which are referred to as the cap structure that is added by RNA polymerase II at early stages in transcription and prior to other RNA modifications [82]. The cap structure comprises a methyl group at position N-7 of the 5′-terminal guanosine nucleotide, designed as m7G. Adjacent nucleotides may undergo methylation at their 2′-O positions, mediated by ribose 2′-O methyltransferase, resulting in three categories of methylation: cap 0, cap 1, and cap 2. Caps 0 and 1 are methylated within the nucleus, while cap 2, characterized by additional 2′-O methylation, proceeds in the cytoplasm. Eukaryotic viruses typically exhibit mRNAs with cap 1 and cap 2 structures, whereas viruses infecting unicellular organisms and plants predominantly utilize cap 0 structures [82].

The cap is essential for mRNA stability, cellular transport, translation initiation, and for preventing immune detection [83]. In the absence of a 5′-cap, RNA is recognized as foreign, triggering an immune response mediated by RIG-I or IFIT proteins [84,85,86]. To counteract this, viruses have evolved various strategies for capping their RNAs. Mononegaviruses possess an RNA-dependent RNA polymerase (L) that contains a methyltransferase (MTase) domain responsible for the addition of a guanosine cap structure (cap-G) to their mRNAs [87,88,89,90]. This mechanism was initially characterized in VSV [91], showing similarities to other viruses within this family, though the precise mechanism remains poorly understood. The capping mechanism involves several steps: the L protein first employs its GTPase domain to hydrolyze GTO into GDP by removing the γ-phosphate. Subsequently, the PRNTase domain transfers a monophosphate to GDP to yield GpppA-RNA. The MTase domain then catalyzes methylation of the 5′-nucleotide to form GpppAm-RNA and subsequently adds a methyl group to the guanine, producing 7mGpppAm-RNA [92,93,94,95,96].

In contrast, segmented negative-sense RNA viruses utilize a distinct capping mechanism, called cap-snatching, by which they hijack the caps of host cellular mRNAs. This involves cleaving the 5′ ends of mRNA caps from host transcripts and incorporating them into their own viral mRNAs, though the specifics vary among viral families [97,98,99,100]. For example, in the influenza virus, which replicates in the nucleus, the PB2 subunit of its RNA-dependent RNA polymerase (RdRp) binds to the 5′ cap of host mRNAs. The PA subunit catalyzes an endonucleolytic cleavage, resulting in a short RNA fragment that is incorporated into the initiation of viral mRNA, also serving as a primer for RNA synthesis. Conversely, Bunyaviruses like Hantavirus replicate in the cytoplasm, with the L protein crucial to binding and cleaving the cap from cellular mRNA. Similar to the influenza virus, this short cap fragment functions as a primer for viral transcription [100].

Regardless of the mechanism, the 5′-cap structure is essential for viral transcription and replication, enabling these viruses to effectively evade immune recognition, particularly from RNA sensors such as MDA5 and RIG-I [100,101,102], and from interferon-induced restriction factors like IFIT1. The IFIT1 protein acts as an antiviral effector within innate immunity, selectively recognizing the 5′ end of the viral mRNA to inhibit its translation [103].

## 7. Other RNA Modifications

In this review, we examine key RNA modifications identified to date, particularly highlighting the potential implications of additional modifications within the framework of negative-sense RNA viruses. A significant case is the C-to-U hypermutation catalyzed by the cytidine deaminase APOBEC3G, which has been shown to hinder viral transcription and protein synthesis in cells infected with measles, RSV, and mumps viruses. Notably, it has been established that only mRNA molecules located outside replication factories interact with APOBEC3G [104]. Additionally, it has been described in the context of measles virus that APOBEC3G modulates host cell expression, adversely affecting viral replication through pathways such as a kinase complex linked to mTORC1, which in turn influences cellular metabolism and ultimately impacts measles virus replication [105].

Despite a lack of clarity regarding the underlying mechanisms, it has been documented in retroviral systems that APOBEC3G is crucial for the antiviral response during the single-strand DNA stage of infection [106,107]. Furthermore, APOBEC3G forms interactions with various cytoplasmic proteins within distinct ribonucleoprotein complexes (RNPs), including processing bodies (p-bodies) and stress granules, both of which play vital roles in RNA metabolism regulation in eukaryotic cells [108,109,110]. This observation raises the intriguing possibility of akin mechanisms being operative in the context of negative-sense RNA viruses.

Additionally, N1-methylpseudouridine, a methylated form of pseudouridine, has emerged as a key modification in in vitro transcription and RNA vaccine development. This modification not only enhances the structural stability of RNA but also boosts translational efficiency, which is critical for inducing robust immune responses in vaccine candidates, as exemplified by the effective development of COVID-19 vaccines [111,112]. The implications of these modifications extend to the design of vaccines targeting negative-sense RNA viruses. However, an initial study involving a vaccine for Andes virus (ANDV) revealed challenges regarding immunogenicity and protective efficacy in murine models [113], highlighting the necessity for further exploration in this domain.

## 8. The Impact of Chemical Modifications on the Genome of Negative-Sense RNA Viruses Beyond the Immune Response

Negative-sense RNA viruses leverage their RNA genomes as multifunctional elements crucial to their replication cycles. The interplay between the primary nucleotide sequence and secondary structural features—including stem-loops and pseudoknots—alongside the viral promoter sequences necessary for polymerase action, and the regulatory regions at the 3′ and 5′ ends of the genome, such as Leader (le) and Trailer (tr), establishes a complex functional architecture vital for viral propagation. RNA modifications are not isolated entities; instead, they are intricately integrated into this structural and functional framework. Importantly, the effects of specific RNA modifications can vary significantly depending on their location within structured domains, their proximity to translation initiation sites, and their interaction with regulatory motifs. This underscores the critical nature of positional context in determining RNA functionality in the viral lifecycle. For example, m6A modification is crucial for transcriptional activation of various genes in influenza A virus (IAV), and the m6A reader YTHDC1 plays a pivotal role in modulating IAV mRNA splicing [31,114]. Furthermore, recent findings indicate that a deficiency of m6A in IAV viral RNA considerably impairs the assembly of the viral ribonucleoprotein complex (vRNP). This deficiency disrupts interactions between viral RNA and vRNP proteins, which rely on the action of m6A methyltransferases [115]. Additionally, prior research conducted by our team indicates that the methylation of METTL3 and the binding of the YTHDF protein may inhibit the coalescence of viral factories [37].

Moreover, single-stranded RNA viruses frequently exploit host-derived RNA modification machinery, indicating that the evolution of these modifications may be intricately linked with the structural features of RNA to enhance the virus–host interaction. This highlights the importance of incorporating RNA folding dynamics, interactions with both viral and host proteins, and the temporal dynamics of infection when analyzing RNA modifications. A comprehensive understanding of these chemical modifications extends beyond merely assessing their roles in immune evasion, providing deeper insights into their overall impact on viral pathogenesis.

## 9. Conclusions and Future Directions

Post-transcriptional modifications of viral RNA are crucial for enhancing viral replication through two primary mechanisms: they increase the stability and translational efficiency of viral mRNA and facilitate evasion of host innate immune responses (see Table 1). This evolutionary pressure has driven viruses to adopt complex strategies for the integration of various epitranscriptomic modifications within their RNA genomes.

Moreover, several modifications remain inadequately characterized due to a lack of empirical data, underscoring an urgent need for deeper investigations into both known and prospective novel chemical modifications and their roles in viral pathogenesis. The advent of advanced sequencing and transcriptomic technologies offers exciting opportunities to elucidate these complex interactions, potentially leading to innovative therapeutic approaches aimed at disrupting viral replication and enhancing host immune responses.

In fact, one of the key limitations of this review is the insufficient data regarding the interactions of various nucleobase modifications and their impact on viral RNA dynamics and the subsequent immune response. Notably, certain modifications, including m6A, m6Am, and inosine, are known to stimulate pro-inflammatory pathways, suggesting potential synergistic or additive effects. However, evidence supporting these hypotheses is currently lacking.

While existing research predominantly relies on cellular models, there is a significant opportunity to establish animal models that can provide a more comprehensive understanding of how these modifications influence the pathogenesis of viral infections. Ultimately, a thorough exploration of the intricate mechanisms underlying negative-sense single-stranded viruses is essential for advancing our knowledge in this area.

## Figures and Tables

**Figure 1 viruses-17-00795-f001:**
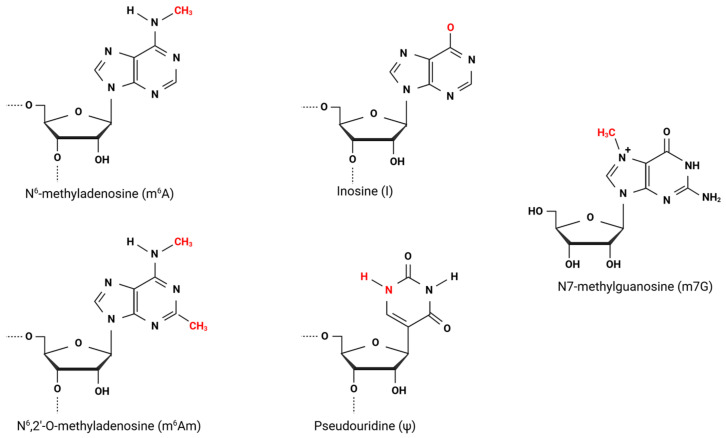
Chemical modifications of RNA. Structural representations of RNA modifications that influence the biology of negative-sense RNA viruses. Created with BioRender.com.

**Figure 2 viruses-17-00795-f002:**
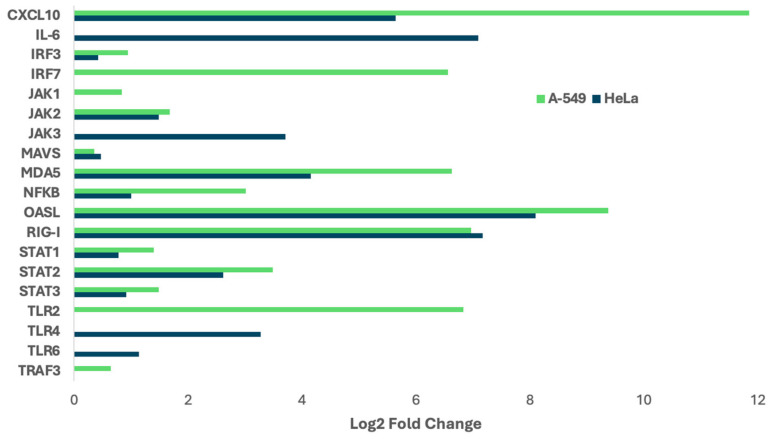
Differential gene expression following RSV infection. A focused analysis of genes associated with the innate immune response during human respiratory syncytial virus infection was conducted using publicly available m6A-sequencing data [32].

**Table 1 viruses-17-00795-t001:** Chemical RNA modifications affecting negative-sense RNA viruses.

Modifications	Enzymes	Virus Described	Role in Innate Immunity
m^6^A	METTL3/14* [15,16] FTO* [19] ALKBH5* [20] YTHDF1-3* [21]	Respiratory syncytial virus [32] Influenza virus [29,31] Sendai virus [27] Measles virus [27] Metapneumovirus [27] Vesicular stomatitis virus [27]	🠯 RIG-I [33] 🠭 IL-6 [33] 🠭 TNFα [33]
m^6^Am	PCIF1* [44,46] FTO* [40]	Vesicular stomatitis virus [45] Rabies virus [46] Measles virus [46]	Pro-inflammatory [47]
Inosine	ADARs* [52]	Respiratory syncytial virus [60] Measles virus [59] Parainfluenza virus [60] Influenza virus [51]	🠭 IL-6 [60] 🠭 IFN-β [60] 🠭 TNFα [60] 🠭 MAPK [60]
Pseudouridine	PUS* [76,77]	Influenza virus [71]	Unknown Evasion? [33]
m7G	L MTase domain** [85,86,87,88] L and PA**	All non-segmented Mononegavirus [89] Influenza virus [95] Hantavirus [96,97,98]	Evasion [81,98,100]

(*) cellular protein; (**) Viral proteins.

## Data Availability

No new data were created or analyzed in this study. Data sharing is not applicable to this article.

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
