# Peer review of "The Role of Chemical Modifications in the Genome of Negative-Sense RNA Viruses on the Innate Immune Response"

_viruses, 2025, doi:10.3390/v17060795_

Round 1
Reviewer 1 Report
Comments and Suggestions for Authors
Manuscript by Ceballos MA and Acevedo ML untitled “Role of Chemical Modifications in the Genome of Negative-Sense RNA Viruses on the Innate Immune Response” presents a cascade of immunological responses related with the presence of several modifications within various RNA viruses. Manuscript could be interesting and published in the Viruses journal after revision.
Below are some comments and suggestions to the authors:
1/ in the manuscript, which in title is term “negative-sense RNA”, there is very little information about viral RNAs. In my opinion, focusing only on selected viral RNA modifications is not sufficient. Modifications are a small part of viral RNA, and paying no attention to the entire RNA is an error. It is related to questions concerning the influence of RNA modifications on biological function, structure and interaction of viral RNA.
2/ in viral RNA is present simultaneously several modified nucleotides. In that case, described in the manuscript immunological responses are the consequence of the presence of single particular RNA modification or it is a response to the presence all viral RNA modifications?
3/ one of the minor modifications of RNA is N1-methylpseudouridine. In Covid-19 related vaccines all uridine residues were replaced with N1-methylpseudouridine because that modification results in very good immunological response. In my opinion, the authors should include that information in the manuscript.
4/ why do the names of sections 4 and 5 of the manuscript contain the word “editing”? All described RNA modifications are introduced/edited posttranscriptionally by respective enzymes as it is also in the case of inosine (ADAR) and pseudouridine (Box H/ACA ribonucleoproteins). Those names of 4 and 5 sections may suggest different ways of introducing N6-methyladenosine and N6,2’-O-methyladenosine into viral RNA.
Comments on the Quality of English Languagenon
Author Response
Comments 1. In the manuscript, which in title is term “negative-sense RNA”, there is very little information about viral RNAs. In my opinion, focusing only on selected viral RNA modifications is not sufficient. Modifications are a small part of viral RNA, and paying no attention to the entire RNA is an error. It is related to questions concerning the influence of RNA modifications on biological function, structure and interaction of viral RNA.
Response 1: I concur with this observation. Consequently, we have incorporated Subtitle 8: “Impact of Chemical Modifications on the Genome of Negative-Sense RNA Viruses Beyond the Immune Response” (line 359 to 390). In this section, we delve into the implications of these modifications and broaden our examination of the topic's relevance within the context of the review.
Comments 2. In viral RNA is present simultaneously several modified nucleotides. In that case, described in the manuscript immunological responses are the consequence of the presence of single particular RNA modification or it is a response to the presence all viral RNA modifications?
Response 2. I concur with the validity of this question. However, addressing this topic proves challenging due to insufficient evidence. This issue is discussed further in Section 9, Conclusions and Future Directions, specifically between lines 404 and 409.
Comments 3. One of the minor modifications of RNA is N1-methylpseudouridine. In Covid-19 related vaccines all uridine residues were replaced with N1-methylpseudouridine because that modification results in very good immunological response. In my opinion, the authors should include that information in the manuscript.
Response 3. We concur with this observation, and the relevant information has been incorporated in Section 7, titled "Other RNA Modifications," specifically within lines 349 to 357.
Comments 4. Why do the names of sections 4 and 5 of the manuscript contain the word “editing”? All described RNA modifications are introduced/edited posttranscriptionally by respective enzymes as it is also in the case of inosine (ADAR) and pseudouridine (Box H/ACA ribonucleoproteins). Those names of 4 and 5 sections may suggest different ways of introducing N6-methyladenosine and N6,2’-O-methyladenosine into viral RNA.
Response 4. We agree with this statement, and it was a mistake to make that distinction; therefore, it has been corrected.

Reviewer 2 Report
Comments and Suggestions for Authors
Review report on manuscript 3571173 submitted to Viruses
Role of Chemical Modifications in the Genome of Negative-Sense RNA Viruses on the Innate Immune Response
Maria-Alejandra Ceballos and Monica L. Acevedo
The authors present here a review on chemical modification in the negative sense RNA viruses and their role on innate immune response. It is clear and well written and the subject is quite relevant.
As major comments, the part concerning adenosine to inosine editing should be more detailed about ADAR proteins : how many ADAR proteins are present in the human proteome (and which ones actually exert A to I activity). There are two isoforms of ADAR1, with the longest one being regulated by IFN.
Furthermore, even little is known about the impact of the cytidine deaminases (APOBEC) on single-stranded negative RNA viruses, a part should be added in this review. Some studies concerning paramyxoviruses (Measles, Mumps and respiratory syncytial viruses) have shwon that APOBEC3G affects their replication for instance.
Minor comments:
- Line 48. To date, over 150 RNA distinct RNA modifications
- Line 147. (…) inducing a strong IFN response in a manner RIG-I-dependent => in a RIG-I-dependent manner.
- Lines 287-288. In contrast, segmented negative sense RNA viruses utilize a distinct mechanism for capping-cap-snatching-whereby they hijack the caps of host cellular mRNAs. => a distinct capping mechanism, called cap-snatching, by which they hijack the caps of host cellular mRNAs
Author Response
Comments 1. As major comments, the part concerning adenosine to inosine editing should be more detailed about ADAR proteins : how many ADAR proteins are present in the human proteome (and which ones actually exert A to I activity). There are two isoforms of ADAR1, with the longest one being regulated by IFN.
Response 1. I concur with this statement; this information is found in section 4, specifically between lines 205 and 210.
Comments 2. Furthermore, even little is known about the impact of the cytidine deaminases (APOBEC) on single-stranded negative RNA viruses, a part should be added in this review. Some studies concerning paramyxoviruses (Measles, Mumps and respiratory syncytial viruses) have shwon that APOBEC3G affects their replication for instance.
Response 2. I concur with this statement, which is found in section 7, "Other RNA modifications," between lines 329 and 347.
Comments 3: Line 48. To date, over 150 RNA distinct RNA modifications
Response 3: The error was corrected based on the reviewer's recommendations provided in line 49.
Comments 4: Line 147. (…) inducing a strong IFN response in a manner RIG-I-dependent => in a RIG-I-dependent manner.
Response 4: The error was corrected based on the reviewer's recommendations provided in 157
Comments 5:; Lines 287-288. In contrast, segmented negative sense RNA viruses utilize a distinct mechanism for capping-cap-snatching-whereby they hijack the caps of host cellular mRNAs. => a distinct capping mechanism, called cap-snatching, by which they hijack the caps of host cellular mRNAs
Response 5: The error was corrected based on the reviewer's recommendations provided in 308-209
Round 2
Reviewer 2 Report
Comments and Suggestions for Authors
The authors improved the manuscript according to my recommendations, providing even a more comprehensive review on chemical RNA modifications affecting negative single stranded RNA virus genomes.